# Management of Weeds in Maize by Sequential or Individual Applications of Pre- and Post-Emergence Herbicides

Harun Alptekin [1,*], Abdullah Ozkan [1], Ramazan Gurbuz [2] and Muhittin Kulak [3,*]

1 Department of Agricultural Science, Postgraduate Education Institute, Igdir University, Igdir 76000, Turkey
2 Department of Herbology, Faculty of Agriculture, Igdir University, Igdir 76000, Turkey
3 Department of Herbal and Animal Production, Vocational School of Technical Sciences, Igdir University, Igdir 76000, Turkey
* Correspondence: harunalptekinn04@gmail.com (H.A.); muhyttynx@gmail.com or muhittin.kulak@igdir.edu.tr (M.K.)

**Abstract:** Weeds impose serious problems in maize (corn) production, resulting in reduced crop yields and lower profits for farmers. The adverse effects of weeds have been attributed to the competition with maize plants for light, water, and nutrients, which can result in reduced growth and yield of the crop. In this context, effective weed management is important to minimize the negative impact of weeds on maize production. This can be achieved through a combination of cultural, mechanical, and chemical control methods. The use of pre-emergence and post-emergence herbicides as sequential or individual applications of these herbicides can be an effective way to manage weeds in maize. Two consecutive field experiments were conducted in 2019 and 2020 in order to determine the efficacy of sequential and individual applications of Dimethenamid-P + Terbuthylazine and Isoxaflutole + Thiencarbazone methyl + Cyprosulfamide as pre-emergence. On the other hand, Mesotrione + Nicosulfuron and Dicamba + Nicosulfuron were used as post-emergence herbicides. The effects of the herbicides were also assayed on corn yield and related parameters. In this regard, we designed the experiment in a randomized complete block design with four replications. Accordingly, the effect of the herbicides varied according to the active ingredients of the herbicide and the weed species. In addition, the effectiveness of herbicides varied according to the assessment times. The lowest effect was found on *E. crus-galli*, which was obtained from Mesotrione + Nicosulfuron (MN) (35%) plots. Other herbicides showed high efficacy (95–100%). Concerning values of both years, the highest cob length, cob diameter, 1000-grain weight, and plant height were obtained in weed-free control plots and the highest grain yield was obtained in the control plots with weed-free checks as 12.88 tons/ha and 12.37 tons/ha, respectively. The lowest corn grain yield was obtained in weedy control plots in both years. Our findings demonstrate that the combination of pre- and post-emergence herbicides in maize can be an effective way chemical weed control option.

**Keywords:** weed management; biotic stress; dry weed biomass; active ingredient

## 1. Introduction

Maize (*Zea mays* L.) is of the ancient and iconic cereals through the world, owing to its wide range of uses such as human food, animal feed, and biofuel (ethanol production) [1]. As a crucial source of food [2], maize is one of the significant oil sources with a rate between 5–8% [3,4]. Maize production in the world has been increasing continuously over the years, reaching to 1,162,352,997 tons in the world in 2020 [5]. In Türkiye, following wheat production, maize production ranks second with an approximate value of 7 million tons [6]. Due to the rapid increase in the world's population, it is necessary to ensure the plant production to meet the nutritional needs [7]. The relevant demand can be ensured with the crop productivity through buffering/alleviating the stress factors available [8]. In addition to the abiotic stress factors, biotic factors also critically suppress the growth

and performance, which are then translated into the reduced yield of crop productivity. Amid the biotic factors, weeds are one of the critical factors causing yield losses because of competition in the fields for light, nutrients, and water [9–12]. In addition to the competition with maize, weeds might also introduce pathogenic bacteria and viruses, which in turn cause critical reductions in yield [13]. As noted by Güncan and Karaca [14], the reported crop loss/reduced crop productivity might be linked to the geographical regions and cultivar/genotypes of the plants. A plethora of annual and perennial weed species has been documented to have negative effects on maize yield [15–29].

According to the report of Oerke and Dehne [30], a 37% reduction in maize production was observed under weed pressure. In this context, weed control must be done at the earlier periods of growth, whether the corn plant is grown for grain or for silage. As a matter of fact, the plant suffers a lot from weeds during its growth stage. In the fields where weeds are dense, the corn plant might not be able to produce sufficient roots. For this reason, any attempts at this stage are prerequites [14]. The major challenge in yielding higher maize productivity is associated with control and management of weed growth [13]. Of the attempts, mechanical control is not suitable for weed management in maize plantations, since it requires a high amount of labor and is not economical [31,32]. In addition, hand weeding and hoeing methods were effective in coping with the annual weeds, but they do not show the same success in controlling perennial weeds. In this regard, it was stated that the use of herbicides for the control of perennial weeds give better results [22,25,29]. Due to high labor and costs in maize cultivation areas, chemical control methods are preferred because of their fast results, easy application, and low cost [29,33]. Herbicides are of the most effective and widespread attempts to cope with the weeds. However, in order to achieve the desired results with herbicide application, the appropriate herbicide should be used at the appropriate time and in the appropriate dose [34]. In addition, the basic principle of controlling weeds is to know the weed species and their biology well [35]. Corresponding to chemical control of weeds, the control of weeds can be completely ensured if the correct identification of weed species, spraying at the right time, choosing the right herbicide, using the right dose, and spraying with the right method [36]. The classification of herbicides is done upon different aspects such as chemical families, site of action, mode of action, translocation, time of application, selectivity, etc. [37–39]. Herbicides are classified as pre-emergence or post-emergence based upon the time of application [40].

In practical, farmers use both pre-emergence and post-emergence herbicides intensively in corn planting areas [41]. Since pre-emergence herbicides are applied to the leaves, the selection of these herbicides should consider the characteristics of each area, weed species, and agro-climatic conditions [28]. Herbicides applied to the soil reduce the weed population as most of the germinating weeds are suppressed [42]. These weeds mostly consist of annual weeds that reproduce by seed. The effect of herbicides applied to the soil lasts about 40–50 days. Following this period, the secondary weed infestation, which requires foliar application, begins [43].

Therefore, it is necessary to control weeds again in the case of secondary weed infestation. Thus, post-emergence herbicides should be used. In post-emergence herbicides, the target is on emerged annual and perennial weeds. Herbicides might exhibit both positive and negative consequences. However, when the combined use of pre-emergence and post-emergence herbicides targets both annual and perennial weeds, it will have more effects on weed. Similarly, Işık et al. [44] reported that pre-emergence and post-emergence herbicides were more effective when used in combination. However, it is worthy to note that the experimental conditions, such as locations, maize cultivar, bioactive compound of herbicides, mode of actions of the herbicides, as well as weed species and their densities, are also critical predictors in weed management. In this context, we hypothesized that the combined effects of pre- and post-emergence herbicides would be higher than their individual effects on weed control in maize plantations. For that reason, we designed the present study to compare the individual and combined effects of the herbicides with the different bioactive ingredients. In order to test the hypothesis, a series of parameters, such

as weed dry weight, grain yield, plant height, kernel rows, cob length, core diameter, and 1000-grain weight, were recorded in maize plants.

## 2. Materials and Methods

### 2.1. Location and Experimental Soil Properties

Field experiments were conducted during two consecutive seasons in 2019 and 2020 under field conditions (Kızıltepe, Mardin, Türkiye; 370958 N-402539 E). The physico-chemical properties of the experimental soils were as follows: salt free (0.0039 mmhos cm$^{-1}$), pH: 7.34, organic matter content (2.37%, medium), high lime (23.94%), phosphorus content (0.4122 P$_2$O$_5$ kg ha$^{-1}$; low), clay-loam, and potassium (16,675 K$_2$O kg ha$^{-1}$, rich). The weather conditions of the experimental location for the period of 2019–2020 were presented in Table 1.

**Table 1.** The weather conditions of the region.

| Months | Temperature (°C) | | | Precipitation (mm) | | | Humidity (%) | | |
|---|---|---|---|---|---|---|---|---|---|
| | 2019 | 2020 | LTP | 2019 | 2020 | LTP | 2019 | 2020 | LTP |
| March | 10.7 | 10.7 | 12.2 | 95.8 | 157.3 | 59.18 | 86.7 | 65 | 69 |
| April | 13.9 | 14.1 | 16 | 79.7 | 51.6 | 37.62 | 94.3 | 59.7 | 63 |
| May | 22.7 | 19.9 | 21.7 | 49.2 | 30.5 | 38.77 | 78.9 | 43.4 | 47 |
| June | 29.5 | 26.2 | 28.5 | 16.3 | 31.5 | 3.53 | 24.0 | 26 | 25.1 |
| July | 30.8 | 31.5 | 32.1 | 1.7 | 4 | 0.73 | 27.03 | 20.6 | 21 |
| August | 31.7 | 29.9 | 30.9 | 0.1 | 0 | 0.2 | - | 22.1 | 27.6 |
| September | 26.3 | 29.3 | 26.2 | 0.3 | 0 | 1.47 | - | 20.6 | 30.5 |
| October | 22.3 | 22.8 | 20.5 | 32.7 | 0 | 24.51 | - | 22.5 | 38.3 |

Data from [45]. Long-term period: LTP.

### 2.2. Plant Material and Experimental Design

The Seeds of DKC6664 (Monsanto, DeKalb, IL, USA) maize variety were used for the study. Two pre-emergence and two post-emergence herbicides were used in the study; namely, Dimethenamid-P + Terbuthylazine (DT), Isoxaflutole + Thiencarbazone methyl + Cyprosulfamide (ITC), Mesotrione + Nicosulfuron (MN), and Dicamba + Nicosulfuron (DN) (Table 2). Briefly, maize seeds were sown as a second crop in every two years on 24 June 2019 and 17 June 2020, respectively. The first product was wheat, and maize was planted after the soil was cultivated with a double-acting disc harrow from the wheat harvest. The seeds (0.25 kg ha$^{-1}$) were sown at an inter-row distance of 70 cm and intra-row distance of 20 cm. In both years of the study, 40 kg of 20 + 20 + 20 (NPK) fertilizers were applied per decare with planting and 35 kg of urea (46% N) per decare 45 days after planting. After planting, the first irrigation was done with the sprinkler irrigation method and the following irrigations were done with the drip irrigation method once per 7 days in the first months and once per 5 days in the last two months. Due to the use of herbicide before emergence in the study, parcellation study was carried out after crop planting.

The experiment consisted of forty plots with ten experimental groups (DT, ITC, MN, DN, DT + MN, DT + DN, ITC + MN, ITC + DN, weed-free, and weedy) and four replications according to the randomized blocks design (Figure 1). The area of each plot is 5 m × 4 m = 20 m$^2$ and the total trial area is 1230.5 m$^2$. The distance between each treatment and replication were as 1.5 m and 1 m, respectively. The herbicides were applied 3 days after sowing (27 June 2019 and 20 June 2020) and post-emergence herbicides 22 days after sowing (16 July 2019 and 9 July 2020). In the study, a back sprayer with a 25-L tank capacity, gasoline engine, and fan nozzles was used for herbicide application. Hoeing and hand plucking was done in the weed-free plots in case of emergence of weeds.

### 2.3. Determination of Weed Species and Densities in the Experimental Area

Prior to the establishment of the trials, weed species and their densities were noted. In this regard, a 1 m$^2$ frame was used in the trial area, randomly replaced, and the weed

species, growth stages, and the number of each weed species in the covered area or m$^2$ were recorded. The densities of each species were calculated according the following equation [46]:

$$\text{Density (plants/m}^2) = B/m,$$

where "B" indicates the total number of individual plants in the samples and "m" represents the total number of meters.

In addition, the scale suggested by Üstüner and Güncan [47] was used to determine the density of the species (Table 3).

**Table 2.** Active ingredients, mode of action, formulations, dose, and applications times of herbicides.

| Treatments | Acronym | MOA * | Formulation ** | Dose | Application Time *** |
|---|---|---|---|---|---|
| Dimethenamid-P + Terbuthylazine | DT | K3, C1 | SE | 30 mL/ha | PreE |
| Isoxaflutole + Thiencarbazone methyl + Cyprosulfamide | ITC | F2 | SC | 3.5 mL/ha | PreE |
| Mesotrione + Nicosulfuron | MN | F2, B | SC | 25 mL/ha | PostE |
| Dicamba + Nicosulfuron | DN | O, B | SG | 2.5 g/ha | PostE |
| Dimethenamid-P + Terbuthylazine + Mesotrione + Nicosulfuron | DT + MN | | | | PreE + PostE |
| Dimethenamid-P + Terbuthylazine + Dicamba + Nicosulfuron | DT + DN | | | | PreE + PostE |
| Isoxaflutole + Thiencarbazone methyl + Cyprosulfamide + Mesotrione + Nicosulfuron | ITC + MN | | | | PreE + PosstE |
| Isoxaflutole + Thiencarbazone methyl + Cyprosulfamide + Dicamba + Nicosulfuron | ITC + DN | | | | PreE + PostE |
| Weed-free control | Weed-free control | | | | |
| Weedy control | Weedy control | | | | |

* MOA: Mode of action; HRAC Mode of Action Classification 2020 (https://hracglobal.com/files/HRAC_Revised_MOA_Classification_Herbicides_Poster.pdf; accessed on 6 January 2023): K3: Inhibition of VLCFAs; C1,2: Inhibition of photosynthesis PS II–Serine 264; F2: Inhibition of HPPD; B: Inhibition of ALS; O: Auxin mimics; ** SE: Suspoemulsion; SC: Suspension concentrate; SG; Water-Soluble Granule; *** PreE: Pre-emergence; PostE: Post-Emergence.

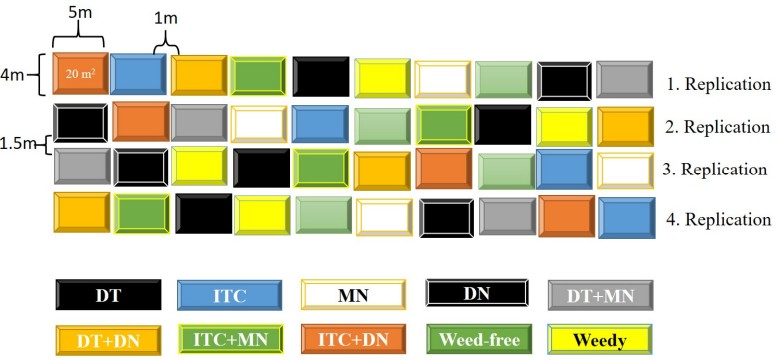

**Figure 1.** Experimental design.

**Table 3.** Density scale of the weeds.

| Scale | Density Level | Density (plants/m$^2$) |
|---|---|---|
| A | High dense | 10+ |
| B | Dense | 1–10 |
| C | Middle dense | 0.1–1 |
| D | Low dense | 0.01–0.1 |
| E | Rare | Less than 0.01 |

### 2.4. Effects of Herbicides on Weed Species and Population

The standard weed herbicide applications were carried out according to the methods of Ministry of Agriculture and Forestry, General Directorate of Agricultural Research and Policies (TAGEM). The percentage of reduction in weed population was determined by comparing the treated plots with the weedy control plots. In order to determine the effects of herbicides on weed population and weed species, changes in weed population and species were observed four times at regular intervals after herbicide applications for the second year [48] (Table 4).

**Table 4.** Assessment and assessment times corresponding to the herbicide treatments.

| Application Time | Assesments | Assessment Intervals |
|---|---|---|
| Pre-Emergence | 1. Assesment | After the completion of the cultivation plant emergence in the control plots |
| | 2. Assesment | 20 days after first assessment |
| | 3. Assesment | At the time when the maize with the tassels |
| | 4. Assesment | Before harvest |
| Post Emergence | 1. Assesment | 10 days after application |
| | 2. Assesment | 25 days after application |
| | 3. Assesment | At the time when the maize with the tassels |
| | 4. Assesment | Before harvest |

Each assessment specifies the phenology of the weeds and the effects on the weeds. The data of the period or periods used as a basis for the biological efficacy evaluation of herbicide were made according to weed populations and individual weed species. Then, the Abbott formula was used for determination of the effect on weeds at the species level and the effects on all weeds [49].

$$\text{Herbicide Percentage effect} = \frac{(\text{Number of Weeds in Control} - \text{Number of Weeds in Treatments}) \times 100}{\text{Number of Weeds in Control}}$$

### 2.5. Effect of Herbicides on Weed Dry Weight

Prior to maize harvest, the weeds found in 1 m$^2$ in each plot were cut from the soil surface separately, put in paper bags, and taken to the Herbology Laboratory. After being kept in an oven at 70 °C for 24 h in the laboratory, they were taken and their dry weights were calculated. Furthermore, the effects of the herbicide weeds were determined based on the weed control plots.

### 2.6. Yield Parameters

Harvest of the plants were done on 23 November 2019 in the first year and on 16 November 2020 in the second year. For assessment of yield parameters, grain yield, plant height, kernel rows, cob length, cob diameter, and 1000-grain weight were recorded in ten maize plants for each replication.

### 2.7. Statistical Analysis

The experimental design corresponded to a factorial model in a completely randomized block, with treatments being herbicide-applied/non-herbicide applied, and weed-free/weed-submitted plants. Considering the number of plants considered for analysis, we have used four replications and each replication corresponded to ten plants. The relevant data were subjected to one-way variance analysis. The means were compared using Duncan's multiple range test ($p < 0.05$) (SPSS 22). In addition, we performed an array of statistics to reduce the dimension and correlate the findings of the study. After transformation/normalization of the data; correlation analysis (JASP), heat map clustering (SR plot), principal component analysis (PAST Software), and network plot analysis (PAST software) were carried out.

## 3. Results and Discussion

### 3.1. Weed Species and Their Density

During the experimental years, a total of 9 weed species belonging to 6 families in the first year and 12 weed species belonging to 8 families in the second year were observed (Table 5). Accordingly, Amaranthaceae (3 species) and Poaceae (3 species) families were of the most observed weeds. The other families were characterized with one species. Corresponding to the leaf structure; 1 of the weed families detected in the experiment area is narrow-leaved and 7 of them are broad-leaved families. Additionally, 3 narrow-leaved weed species and 9 broad-leaved weed species were identified. We observed three perennial and 9 annual species in the experimental area (Table 5). Previously, *A. retroflexus*, *C. album*, *A. theophrasti*, *C. arvensis*, *P. oleracea*, *S. halepense*, and *E. crus-galli* were the weed species observed in maize plantations [15,19,21,24,26,28,50,51], which are consistent with findings of the present study.

**Table 5.** Weed species, families, scientific names, common names, life cycles, and average densities (number/m$^2$) and density level (grade) in the experimental area in both years (2019–2020).

| Family | Scientific Name | Common Name | Life Cycle | 2019 | 2020 | Average |
|---|---|---|---|---|---|---|
| Narrow leaf | | | | | | |
| Poaceae | *Echinochloa crus-galli* (L.) P. Beauv. | Barnyardgrass | A | 3.25 B | 3.20 B | 3.17 B |
| | *Setaria* sp. | Foxtail | A | - | 0.75 C | 0.75 C |
| | *Sorghum halepense* (L.) Pers. | Johnsongrass | P | 10.5 A | 9.00 B | 9.75 B |
| Broadleaf | | | | | | |
| Amaranthaceae | *Amaranthus blitoides* S. Watson | Prostrate pigweed | A | 1.75 B | 2.00 B | 1.87 B |
| | *Amaranthus retroflexus* L. | Redroot pigweed | A | 5.25 B | 4.90 B | 5.07 B |
| | *Chenopodium album* L. | Common lambsquarters | A | 2.30 B | 2.10 B | 2.20 B |
| Asteraceae | *Xanthium strumarium* L. | Cocklebur | A | - | 0.85 C | 0.85 C |
| Brassicaceae | *Sinapis arvensis* L. | Wild mustard | A | 1.75 B | 1.50 B | 1.62 B |
| Convolvulaceae | *Convolvulus arvensis* L. | Field bindweed | P | 2.40 B | 3.20 B | 2.82 B |
| Fabaceae | *Prosopis farcta* (Banks & Sol.) | Syrian mesquite | P | - | 0.75 C | 0.75 C |
| Malvaceae | *Abutilon theophrasti* Medicus | Velvetleaf | A | 0.75 C | 0.50 C | 0.62 C |
| Portulacaceae | *Portulaca oleracea* L. | Purslane | A | 1.15 B | 1.00 B | 1.07 B |

Life Cycle—A: Annual, P: Perennial; A = High density = >10.00 m$^2$, B = Intensive = 1.00–10.00 m$^2$, C = Medium = 0.10–1.00 m$^2$.

With respect to the weed density, in this study, the highest density of *S. halepense* (first year: 10.5 plants/m$^2$, second year: 9.00 plants/m$^2$) was observed in the experimental area in both years. This is followed by *A. retroflexus* (first year: 5.25 plants/m$^2$; second year: 4.90 plants/m$^2$) and *E. crus-galli* (First year: 3.25 plants/m$^2$; second year: 3.20 plants/m$^2$), followed by weed species. Being very similar and consistent with the current study, Uysal Şahin and Kadıoğlu [32], in the first location, reported 9 weed species with a range of 1.1 plant per m$^2$ to 20.1 plant per m$^2$. Amid the weeds observed, the highest density of weed species were found as *A. retreflexus* (20.1 plants/m$^2$), *X. strumarium* (15.6 plants/m$^2$), and *S. halepense* (10.2 plants/m$^2$). In the second location, the highest density was found as *P. oleracea* (18.5 plants/m$^2$), *A. retreflexus* (8.6 plants/m$^2$), and *C. arvense* (8.5 plants/m$^2$). Açıkgöz [51] determined the highest density of weed species of *S. halepense* (37.54 plants/m$^2$), *S. verticillata* (18.95 plants/m$^2$), and *X. strumarium* (9.16 plants/m$^2$) in the trial area. The weed density and identified weed species are very similar to the findings of the present study. In addition, the similar weed species were also reported by Işık et al. [44]., Hançerli and Uygur [21], and Arslan [50]. As for other living organisms, weeds are also critically responsive to the environmental fluctuations, either in biotic or abiotic nature [52,53].

Regardless of environmental conditions, the widespread and density of the weeds might be linked to the characteristics of the species, such as competition ability, seed production, number of seeds per plants, dissemination, and reproduction system (annual and perennial) [52,53].

*3.2. Effects of Herbicides on the Weed Species and Population*

Except first assessment ($p = 0.105$), other assessments and herbicides affected the weed population ($p < 0.05$). In general, the effect on the weed population was higher in the plots with the herbicides used at both application times. Specifically, we observed the highest effect was in DT + MN (35%) plot in the first assessment, ITC + MN (65%) plot in the second assessment, DT + MN and ITC + MN (80%) in the third assessment, and DT + MN with 92% in the final assessment (Figure 2).

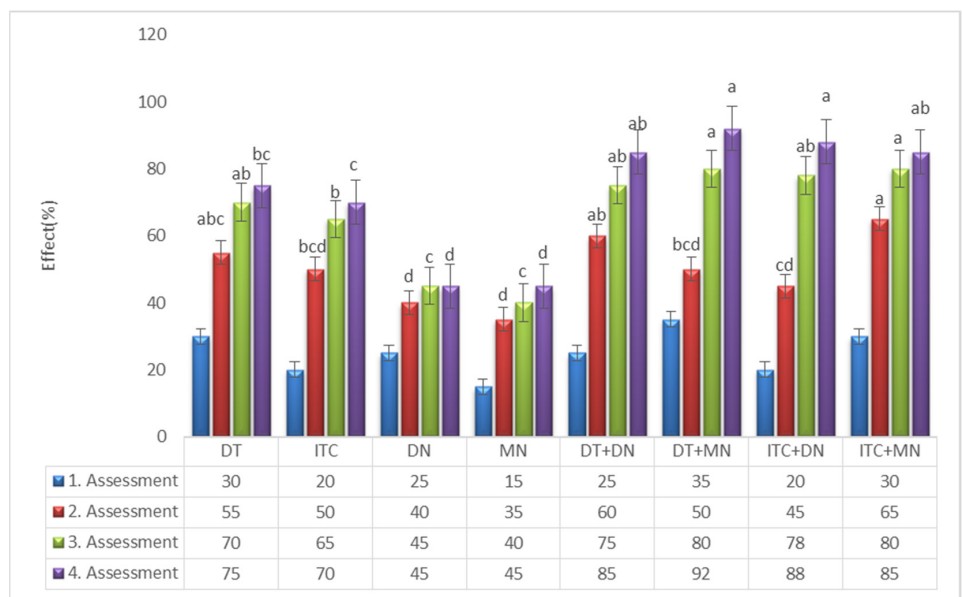

**Figure 2.** Assessment intervals after herbicide applications. Each assessment was compared within itself. According to the variance analysis, the effects of treatments were not statistically significant after first assessment. The differences between the means with the same letter are not significant at the 0.05 level.

As a result of the study, the lowest effect rates were determined in DN and MN plots. In addition, lower effects were determined in plots where herbicides were used at one time. As clearly documented, the effects are dependent on weed species, active compounds of the herbicide, and assessment intervals after the application [24,32,54,55]. Concerning assessments at intervals after the application, the effect rates increased by time.

The effect of the herbicides varied according to the bioactive ingredients of the herbicide, the weed species and assessment intervals. In the following parts of the present study, the weed species (*P. farcta* and *A. theophrasti*) which are not affected by the herbicides were not included.

The lowest effect on *E. crus-galli* was obtained from MN (35%) plots. Other herbicides showed high efficacy (95–100%) (Figure 3A). The lowest effects on Setaria sp. were observed in MN and DN plots, whereas a 100% effect was recorded in other plots (Figure 3B).

The effects of herbicide on *S. halepense* ranged between 90–100% for MN, DN, DT + MN, DT + DN, ITC + MN, and ITC + DN plots. No significant effects were observed in other plots (Figure 3C). Considering *A. blitoides*, effect rates varied between 25% and 100%, and the highest percentage effect was noted in ITC + DN plots (Figure 3D). In the last assessment of *A. retroflexus*, the lowest effect was observed in MN plots with a rate of 45%. In other plots, effect rates of more than 50% were observed (Figure 4A). All herbicides used against *C. album* showed a high effect of 95% and 100% (Figure 4B). In the assessment *of X. strumarium*, the highest effects were observed in MN, ITC + MN, and DT+MN plots as 90%, 95%, and 100%, respectively (Figure 4C). In other plots, the effect rates varied between 10% and 40%. The effect rates of the herbicides on *S. arvensis* in the last assessment varied between 5% and 30% (Figure 4D). While the effects of *C. arvensis* were observed

between 90% and 95% in three plots, low effect rates (5–10%) were observed in other plots (Figure 5A). Among the herbicides used in the last assessment, *P. oleracea* had the lowest effect with a rate of 10% at MN plot (Figure 5B). The effects of other herbicides were found to vary between 90% and 100%. Those values are similar to the former report of Mitkov et al. [24] and Uysal Şahin and Kadıoğlu [32]. As we reported here, the responses of the weeds to herbicides were species specific [56].

The herbicides are classified into two major groups according to their translocation characteristics in plants, i.e., contact and systemic herbicides [57]. As clearly demonstrated by [58,59], the time required for systemic herbicides is related to quite a number of factors, such as herbicide type, target weed, environmental conditions, and application methods. The effects of systemic herbicides might be manifested from a few hours to several weeks. Additionally, the final effects might not be observed for several weeks as the plant continues to absorb the herbicide.

### 3.3. Variance Analysis of Weed Dry Weight and Agronomic Attributes

Either individual or combined pre- and post-emergence treatments critically affected weed dry weight (F = 22.03; $p < 0.01$; F = 59.17; $p < 0.01$ for 2019–2020, respectively) (Table 6). Measuring the dry weight of weeds are considered critical indicators of herbicide efficiency. For that reason, by comparing the dry weight of weeds before and after treatment, it is possible to determine how much the herbicide reduces the weight of the weeds. This can be an indicator of the herbicide's effectiveness at controlling the weeds. Measuring dry weight can also be useful for research purposes. For example, the effects of different environmental conditions on weed growth might be linked to the dry weight of the weeds [55].

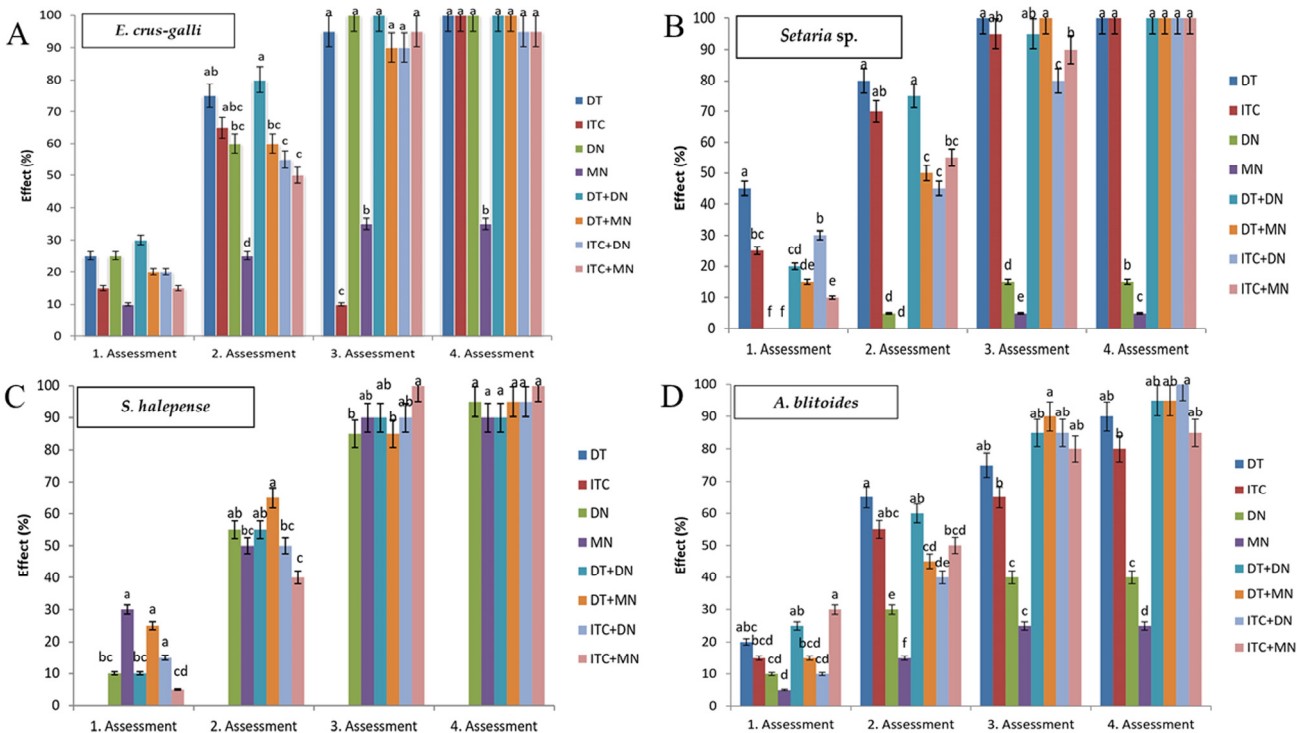

**Figure 3.** Effect of herbicides on (**A**) *E. crus-galli*, (**B**) *Setaria* sp., (**C**) *S. halepense*, and (**D**) *A. blitoides* at different intervals. The differences between the means with the same letter are not significant at the 0.05 level.

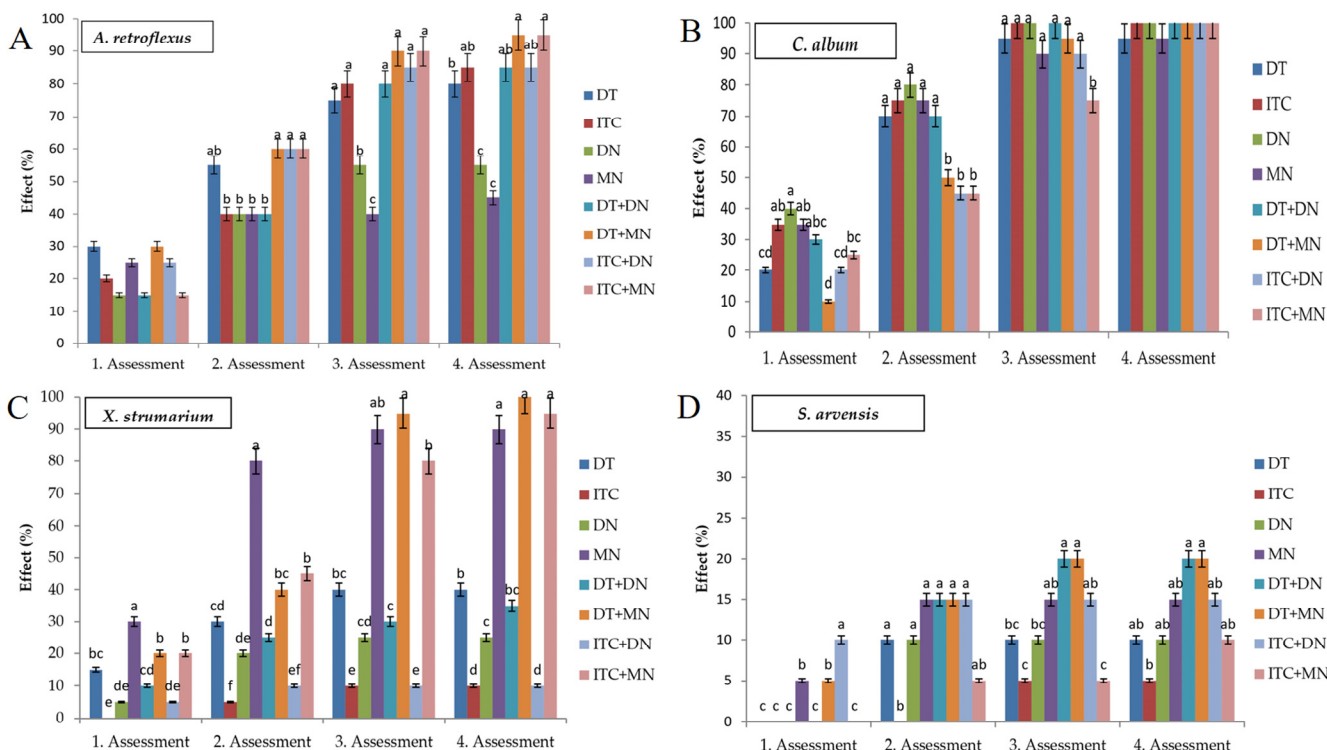

**Figure 4.** Effects of herbicides on (**A**) *A. retroflexus*, (**B**) *C. album*, (**C**) *X. strumarium*, and (**D**) *S. arvensis* at different intervals. The differences between the means with the same letter are not significant at the 0.05 level.

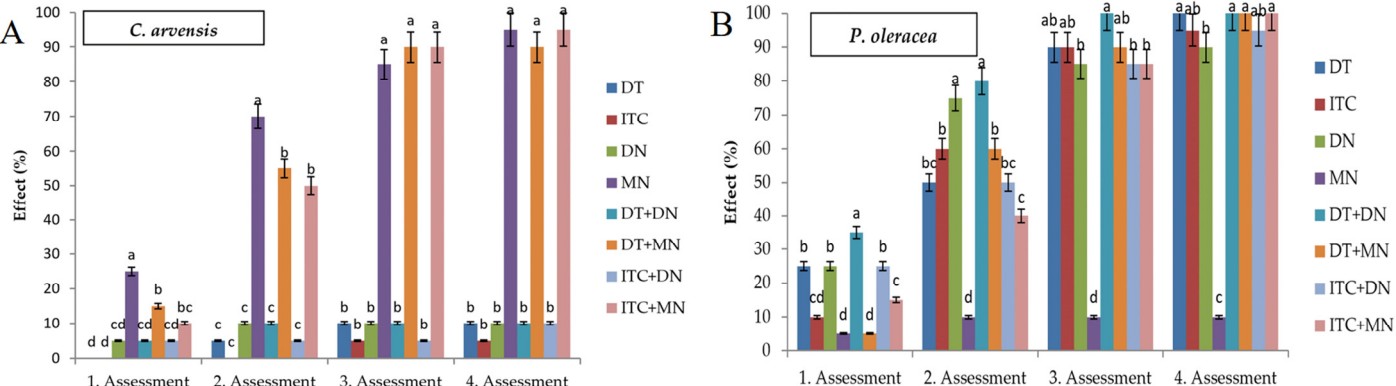

**Figure 5.** Effect of herbicides on (**A**) *C. arvensis* and (**B**) *P. oleracea* at different intervals. The differences between the means with the same letter are not significant at the 0.05 level.

Corresponding to the values of weed dry weight in the first year, the lowest values of weed dry weight were, except for weed-free plots, obtained in the DT (44.12 g m$^{-2}$) and DT + MN (45.02 g m$^{-2}$) plots. As expected, the highest values were recorded (466.25 g m$^{-2}$) in the weedy plot, followed by DN (396.25 g m$^{-2}$) and MN (275.00 g m$^{-2}$). Dry weight values of the second year were as follows: DT (66.15 g m$^{-2}$), DT + DN (69.25 g m$^{-2}$), ITC + DN (71.62 g m$^{-2}$), weedy (406.60 g m$^{-2}$), DN (349.62 g m$^{-2}$), and MN (332.60 g m$^{-2}$). With the respect to the average values of both years, the lowest values were recorded at treatment of DT + MN (71.83 g m$^{-2}$) and ITC + DN (71.97 g m$^{-2}$) (Table 6). Those findings are consistent with the previous reports [17,18,22,25,27,56], indicating that herbicide critically reduced the weed dry weight.

**Table 6.** Variance analysis and effects of treatments on weed dry weight.

| Treatments | 2019 | | 2020 | | Average | |
|---|---|---|---|---|---|---|
| | Dry Weight (g m$^{-2}$) | Effect (%) | Dry Weight (g m$^{-2}$) | Effect (%) | Dry Weight (g m$^{-2}$) | Effect (%) |
| Weed-free | 0.00 d | 100 | 0.00 d | 100 | 0.00 d | 100 |
| DN | 396.25 a | 15.01 | 349.62 b | 14.01 | 372.93 b | 14.54 |
| DT | 44.12 cd | 90.53 | 66.15 c | 83.73 | 55.135 cd | 87.36 |
| DT + DN | 98.75 cd | 78.82 | 69.25 c | 82.96 | 84c | 80.75 |
| DT + MN | 45.02 cd | 90.34 | 98.65 c | 75.73 | 71.83 c | 83.54 |
| ITC | 117.00 c | 74.90 | 115.25 c | 71.65 | 116.12 c | 73.39 |
| ITC + DN | 72.32 cd | 84.48 | 71.62 c | 82.38 | 71.97 c | 83.50 |
| ITC + MN | 73.07 cd | 84.32 | 78.25 c | 80.75 | 75.66 c | 82.66 |
| MN | 275.00 b | 41.01 | 332.60 b | 18.19 | 303.8 b | 30.38 |
| Weedy | 466.25 a | 0.00 | 406.60 a | 0.00 | 436.42 a | 0.00 |
| Mean | 158.78 | 65.94 | 158.80 | 60.94 | 158.79 | 63.61 |
| F | 22.03 | | 59.17 | | 62.83 | |
| *p*-value | 0.00 | | 0.00 | | 0.00 | |
| R$^2$ | 0.93 | | 0.91 | | 0.95 | |

The differences between the means with the same letter are not significant at the 0.05 level.

Regarding effects of herbicide on weeds in the first year, the highest effects were obtained at the DT plot, with 90.53%, and DT + MN, with 90.34%, in 2019, while the lowest effect was recorded at the DN plot (15.01%). In the second year, the highest effects were noted at the DT (18.73%) and DT + DM (82.92%) plots, whereas the lowest percentage was recorded at the DN plot (14.01%). Considering the mean values of both years, the highest and lowest effects were found to be 87.38% at the DT plots and 14.55% at the DN plots, respectively. However, the relevant effects significantly varied according to the herbicides, doses, and weeds [27,29,32]. In addition, the effects of the herbicides on the weeds in the corn fields differed, but all herbicides had exhibited significant alleviating effects on weeds than the weed control plots [56].

In addition, Duncan's multiple comparison test was used to determine the effect of herbicides with different bioactive ingredients on corn cob length, cob diameter, 1000-grain weight, plant height, kernel rows, and grain yield for two years. Accordingly, all parameters were significantly affected by the treatments (*p* < 0.05) (Table 7).

Cob lengths varied between 14.79 and 20.80 cm and 11.65 and 19.62 in the first and second experimental years, respectively. As expected, we obtained the highest values of cob lengths (2019: 20.80; 2020: 19.62 cm) at the weed-free plot. On the other hand, the lowest values of cob lengths were found to be 14.79 cm (2019) and 11.65 cm (2020) at weedy plots. Being very similar to the works of Uysal Şahin and Kadıoğlu [32], the highest and lowest cob lengths were observed as 17.7 cm at the weedy plot and 21.3 cm at the weed-free plot. On the other hand, Açıkgöz [51] reported similar values as 15.24 cm at weedy and 21.2 cm at weed-free plots. Regarding cob diameter, the widest diameters were observed at the weed-free plots (52.69 mm) in the first year and 55.25 cm at the ITC + DN plots. On the other hand, the narrowest values were 45.27 mm and 37.75 mm at in the first and second year, respectively. Those values are consistent with the reports of Açıkgöz [51] and Uysal Şahin and Kadıoğlu [32]. The length of corn cobs might be related to the effectiveness of herbicides for weed control in corn plantations [60,61]. Suspended weed density through herbicide applications were translated to the higher cob length and subsequently to the higher number of grains per cob. The improved attributes of cob were considered to be the consequences of less weed competition and enhanced capability of corn to generate more photosynthetic assimilate via uses of nutrients available [62].

The highest values of 1000-grain yield were found to be 401.95 g, 332.62 g, and 332.37 g at weed-free, DT + MN, and DT plots, respectively, in the first year. On the other hand, the highest values were 381.52 g, 348.75 g, and 338.32 g at weed-free, ITC + DN, and DT + DN plots, respectively, in the the second year. Being consistent with other parameters considered for analysis, the lowest values at the weedy plot were 279.10 for the first year and 268.00 for the second year. Those findings are consistent with the reports of Eymirli

and Uygur [63] and Açıkgöz [51]. The efficacy of herbicides on corn grain yield might vary depending on quite a number of factors, including the type of herbicide being used, the timing of application, the target weeds, and the stage of growth of the maize plants. Herbicides can have both positive and negative effects on maize grain yield [64]. In some cases, herbicides can effectively control weeds that compete with maize for resources, such as water, nutrients, and light, leading to higher grain yields. In other cases, herbicides may cause unintended injury to the maize plants, leading to reduced grain yield [65].

**Table 7.** Variance analysis of agronomic attributes of maize corresponding to the treatments.

| Treatments | Cob Length (cm) | | | Cob Diameter (mm) | | | 1000-Grain Weight (g) | | |
|---|---|---|---|---|---|---|---|---|---|
|  | **2019** | **2020** | **Mean** | **2019** | **2020** | **Mean** | **2019** | **2020** | **Mean** |
| DN | 16.60 f | 15.82 de | 16.21 e | 47.97 d | 45.00 d | 46.48 d | 298.40 bc | 305.97 f | 302.18 cd |
| DT | 17.88 cde | 17.50 bc | 17.69 cd | 52.11 ab | 49.25 cd | 50.68 bc | 332.37 b | 322.60 de | 327.48 bc |
| DT + DN | 18.11 cd | 18.17 abc | 18.14 bcd | 51.02 abc | 52.00 ab | 51.51 ab | 322.35 b | 338.32 c | 330.33 bc |
| DT + MN | 19.81 ab | 16.90 cd | 18.35 bc | 50.21 bc | 51.75 ab | 50.98 ab | 332.62 b | 324.37 d | 328.49 bc |
| ITC | 17.65 def | 17.12 bcd | 17.38 d | 50.81 abc | 48.50 cd | 49.65 bc | 319.00 bc | 315.45 e | 317.22 bcd |
| ITC + DN | 19.01 bc | 18.60 ab | 18.80 b | 49.37 cd | 55.25 a | 52.31 ab | 315.02 bc | 348.75 b | 331.88 b |
| ITC + MN | 19.78 ab | 17.95 bc | 18.86 b | 49.28 cd | 52.00 ab | 50.64 bc | 328.12 b | 331.42 cd | 329.77 bc |
| MN | 16.67 ef | 14.87 e | 15.77 e | 51.20 abc | 45.00 d | 48.1 cd | 324.82 b | 291.37 g | 308.09 d |
| Weed-free | 20.80 a | 19.62 a | 20.21 a | 52.69 a | 54.50 a | 53.59 a | 401.95 a | 381.52 a | 391.73 a |
| Weedy | 14.79 g | 11.65 f | 13.22 f | 45.27 e | 37.75 e | 41.51 e | 279.10 c | 268.00 h | 273.55 e |
| Mean | 18.11 | 16.82 | 17.47 | 49.99 | 49.10 | 49.55 | 325.37 | 322.78 | 324.08 |
| F | 19.60 | 21.18 | 43.955 | 13.34 | 12.39 | 17.00 | 5.28 | 114.28 | 18.86 |
| P | 0.00 | 0.00 | 0.000 | 0.00 | 0.00 | 0.00 | 0.00 | 0.00 | 0.00 |
| $R^2$ | 0.94 | 0.88 | 0.930 | 0.87 | 0.85 | 0.84 | 0.85 | 0.91 | 0.85 |
| Treatments | Maize Plant Height (cm) | | | Maize Grain Yield (kg ha$^{-1}$) | | | Kernel Rows | | |
|  | **2019** | **2020** | **Mean** | **2019** | **2020** | **Mean** | **2019** | **2020** | **Mean** |
| DN | 264.95 c | 254.32 a | 259.63 d | 1.197.50 bc | 1.112.50 f | 1155.00 c | 16.6 d | 18 b | 18.00 b |
| DT | 277.20 b | 273.35 a | 275.27 b | 1.224.50 abc | 1.182.00 de | 1203.25 b | 17.88 c | 19 a | 19.00 ab |
| DT + DN | 275.85 b | 267.92 a | 271.88 b | 1.235.00 abc | 1.209.25 bc | 1222.12 b | 18.11 c | 17 c | 17.50 bc |
| DT + MN | 278.35 ab | 269.82 a | 274.08 b | 1.252.50 ab | 1.186.00 d | 1219.25 b | 19.81 b | 18 b | 18.00 b |
| ITC | 281.45 ab | 264.37 a | 272.91 b | 1.242.50 abc | 1.172.50 e | 1207.50 b | 17.65 | 17 c | 17.50 bc |
| ITC + DN | 277.10 b | 269.80 a | 273.45 b | 1.256.25 ab | 1.217.50 b | 1236.87 ab | 19.01 bc | 18 b | 18.00 b |
| ITC + MN | 277.22 b | 276.42 a | 276.82 b | 1.266.25 ab | 1.205.00 c | 1235.62 ab | 19.78 b | 17 c | 17.00 bc |
| MN | 279.45 ab | 253.92 a | 266.68 c | 1.182.50 c | 1.098.00 g | 1140.25 c | 16.67 d | 19 a | 19.00 ab |
| Weed-free | 288.25 a | 282.40 a | 285.32 a | 1.288.75 a | 1.237.50 a | 1263.12 a | 20.8 a | 19 a | 19.50 a |
| Weedy | 238.25 d | 215.45 c | 226.85 e | 812.50 d | 771.00 h | 791.75 d | 14.79 e | 14 d | 15.00 d |
| Mean | 273.80 | 262.77 | 268.29 | 1.195.82 | 1.139.12 | 1167.48 | 18.11 | 17.6 | 17.85 |
| F | 17.47 | 3.44 | 71.47 | 41.865 | 12.84 | 157.063 | 27.46 | 8.73 | 11.623 |
| P | 0.00 | 0.01 | 0.00 | 0.00 | 0.00 | 0.00 | 0.00 | 0.00 | 0.00 |
| $R^2$ | 0.90 | 0.88 | 0.96 | 0.97 | 0.93 | 0.98 | 0.82 | 0.87 | 0.78 |

The differences between the means with the same letter are not significant at the 0.05 level.

The plant height ranged from 238.25 to 288.25 cm in the first year and ranged between 215.45 cm and 282.40 cm in the second year. The highest values were 288.25 cm (weed-free plot) and 278.35 cm (DT + MN plot) for the first year and 282.40 cm (weed-free plot) and 276.42 cm (ITC + MN plot) for the second year. On the other hand, the shortest plant heights were recorded at the weedy plot as 238.25 cm and 215.45 cm, respectively. Those findings are consistent with the report of Açıkgöz [51].

Weed density can have an impact on the height of maize plants. High weed density can reduce the availability of resources such as water, nutrients, and light for the maize plants, leading to reduced plant height. On the other hand, low weed density might allow maize plants to grow taller because they have access to more resources [66].

The highest values of grain yield were noted at weed-free (12.88-ton ha$^{-1}$), ITC + DN (12.56-ton ha$^{-1}$), and DT + MN (12.52-ton ha$^{-1}$) plots for the first year, as well as weed-free (12.37-ton ha$^{-1}$), ITC + DN (12.17-ton ha$^{-1}$), and ITC + MN (12.09-ton ha$^{-1}$) plots. The lowest values were recorded at weedy plots, i.e., 8.12-ton ha$^{-1}$, and 7.71-ton ha$^{-1}$ in the first and second year, respectively. Very similar reductions were also reported by Eymirli and Uygur [58] and Açıkgöz [51]. In general, a nearly 40% reduction in yield for maize was

reported [14,30,67]. As reported by [68], negative correlations were noted between grain yield of corn and above ground dry weight of weed. As also discussed in the section of multivariate statistical analysis of the present study (Figures 6 and 7), dry weed biomass was negatively correlated to all the attributes of corn, with coefficients in a range of −0.509 to −0.937.

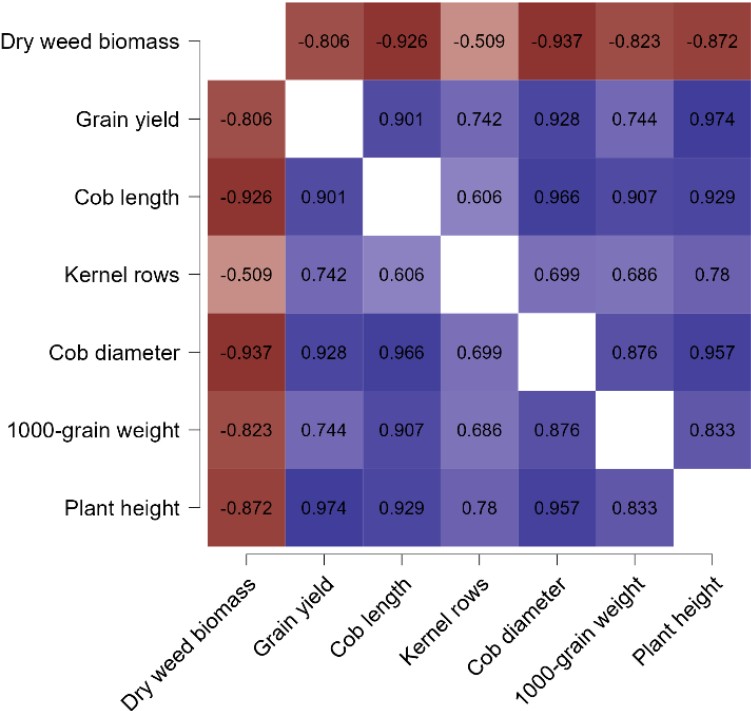

**Figure 6.** Correlation analysis of the estimated parameters.

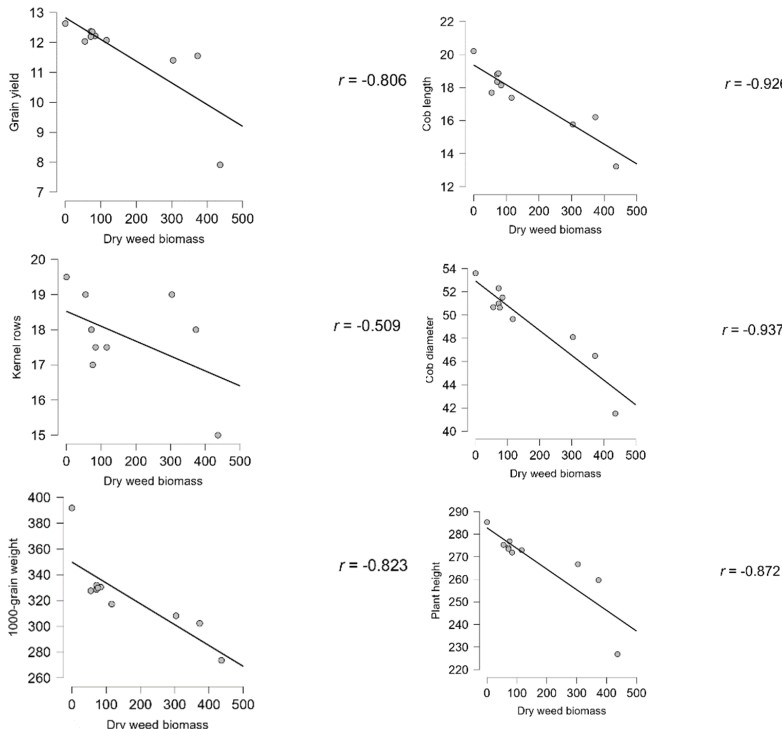

**Figure 7.** Dry weight and other parameter correlations.

*3.4. Multivariate Analysis of the Parameters and Treatments*

In addition to the one-way variance analysis, the mean values obtained were subjected to a series of statistical analysis for reducing the dimension, correlation, visualization, and clarification of the estimated parameters corresponding to the independent treatments. A plethora of documents clearly confirmed the power of multivariate statistical tools in reporting the core parameters of a study, being very common in the case of a high number of dependent/independent variables. As noted above, the present study is a field-based study carried out during period of 2019–2020. For that reason, we performed both individual and combined analysis of the trial year. In this context, herein, we first reported the correlation coefficient between the variables. Since the weed density and its biomass are the critical issues considered in agricultural/non-agricultural fields, we addressed our specific comments on dry weed biomass and its relation to the other variables. According the analysis of 2019, as expected, dry weed biomass was negatively correlated with grain yield ($r = -0.789$ **, $p = 0.007$), cob length ($r = -0.890$ **, $p < 0.001$), core diameter ($r = -0.782$ **, $p = 0.008$), 1000-grain weight ($r = -0.724$ *, $p = 0.018$), and plant height ($r = -0.829$ **, $p = 0.003$) (Supplementary Figure S2). However, no significant correlation with kernel rows was noted ($r = -0.527$ ns, $p = 0.117$). On the other hand, the individual analysis of the second year furthermore supported the individual analysis of the first year (Figures S1 and S2). Considering the combined analysis of the variables considered, we observed consistent coefficients linked to the significance and its direction (Figures 6 and 7).

For that reason, we addressed our further analysis, such as heat map clustering, principal component analysis, and network plot analysis, on the average values of the variables. The analysis values/discrimination/scattering of the individual analysis were presented as Supplementary Materials. In addition, heat map clustering clearly discriminated the dependent/independent variables by sorting them into two major clusters, with a color range (+4 to −4; red to blue) indicating the values obtained (Figure 8). Amid the major clusters, the first cluster included the "weedy" group characterized with the highest values of "dry weed biomass" and lowest values of agronomic attributes including "grain yield, cob length, kernel rows, core diameter, 1000-grain weight, and plant height". This cluster might be considered the "positive control group" and "severe stress groups", respectively. For that reason, the observed values were of the predicted values, according to our best field surveys and to the great number of reports available. We have already designed the present study on potential practical managements to be effective in alleviating the severe pressure/stress of weeds on the plants. The findings of heat map clustering revealed that any attempts here were partially effective in fighting weed, since other treatments and the "control group" were clearly discriminated from the "weedy group".

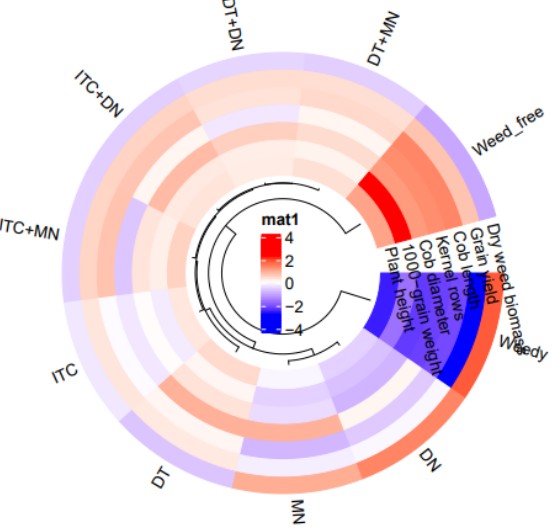

**Figure 8.** Heat map of the parameters corresponding to the treatments.

To consolidate the effects of the attempts/treatments on corn plants, furthermore, we performed a network plot analysis to ascertain the link between individual treatments based on their effects/performance on the agronomic attributes and especially on dry weed biomass (Figure 9). The nodes via lines correspond to the degree of relations, meaning that the thinner/lighter line and thicker line depicts the weaker and stronger relations with each other, respectively. Being consistent with the heat map clustering, a clear discrimination of the "weedy group" was revealed. On the other hand, as expected, other experimental groups, to an extent, were related to each other. In order to determine how and to what degree the groups were similar to each other, furthermore, we consolidated the similarity levels of the group by similarity indices, as inserted on the nodes of the plot.

In order to explain the ratio of variation, agronomic attributes and dry weed biomass were scattered on a biplot pair (Figure 10). Accordingly, two first components (PC1:85.80% and PC2:8.21%) accounted for 94.01% of the variability of the original data. Such a high explained variance clearly suggests that the principal component analysis might successfully be employed in assessing the response of the estimated parameters along with the treatments. The first component (PC1) is positively correlated with all treatments, except groups of DN plots (with a score of −1.61) and weedy plots (with a score of −5.88), including all parameters, except "dry weed biomass" and "kernel rows", which were found to be positively associated with the second component (PC2).

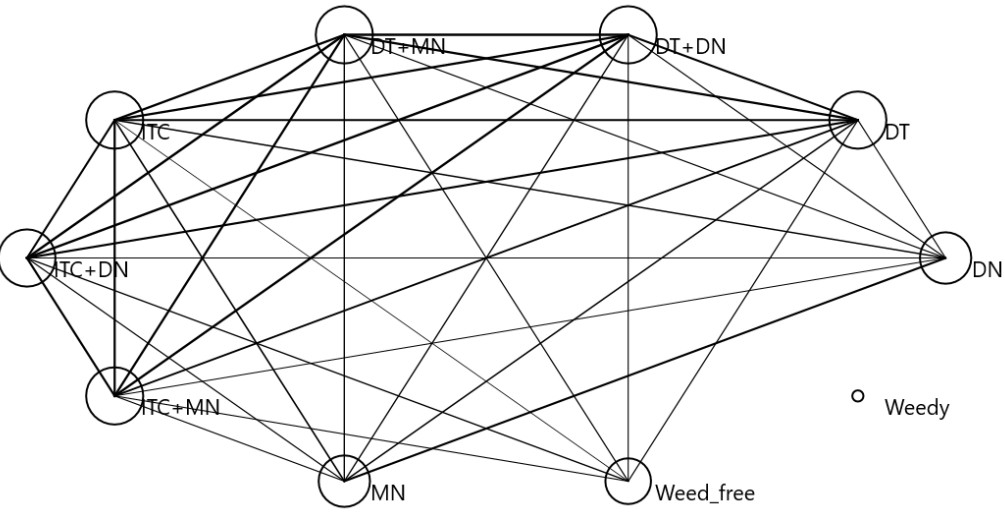

**Figure 9.** Network plot analysis of the treatments.

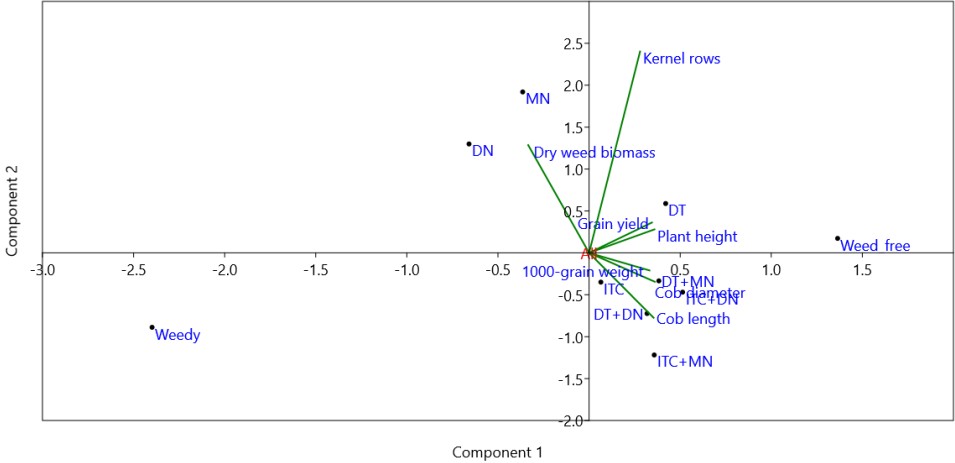

**Figure 10.** Principal component analysis of the parameter and treatments.

Multivariate statistical tools also revealed the critical effects of weeds on growth and productivity of corn, as was clearly reported for other crop species [69–71], also clearly separating the weedy experimental groups from other experimental groups. Such further analysis supports the variance analysis and are considered to be very powerful in reducing the dimension of the variables considered for analysis by clustering the findings. Overall, we clearly noted the effects of herbicides in weed control. We should underline that the effects of herbicides on maize kernel row may be related to the timing of herbicide application and the specific herbicide being used [71].

### 4. Conclusions

In this study, we conducted a study on the potential management of weeds in maize by sequential or individual applications of pre- and post-emergence herbicides. In this regard, a two-year field study was carried out in the southern region of Türkiye. Firstly, we noted a total of twelve weed species belonging to eight families (one narrow-leaved and seven broad-leaved families). Concerning the identified species, we found the common weeds observed in corn plantations, i.e., *A. retroflexus*, *C. album*, *A. theophrasti*, *C. arvensis*, *P. oleracea*, *S. halepense*, and *E. crus-galli*. As expected, and very consistent with former reports, in both experimental years, lower weed dry weights were obtained in the herbicide applied plots in comparison to the weedy plots. For instance, the highest effects on suppressing the weed were observed at DT plots (90.53%) and DT (83.73%) and DT + DM (82.92%) plots, in the first and second year, respectively. Along with suppression of weeds, agronomic attributes of corn were positively affected, as can be seen from correlation coefficients and phenotypes of the plants. In addition, agronomic attributes were positively correlated each other. Significantly, weed-induced reductions of grain yield were buffered with the herbicides. The highest corn grain yield in the first year was found in weed-free control (12.88-ton ha$^{-1}$) and ITC + DN plots (12.56-ton ha$^{-1}$) and the highest weed-free (hoe) (12.37-ton ha$^{-1}$) in the second year was obtained in ITC + DN (12.12-ton ha$^{-1}$) plots. The lowest corn grain yield was obtained in weedy plots in both years. Finally, we might conclude that it might be necessary to use a combination of different methods, including both pre- and post-emergence herbicides, to effectively control weeds and maximize crop yields.

**Supplementary Materials:** The supporting information can be downloaded at: https://www.mdpi.com/article/10.3390/agriculture13020421/s1.

**Author Contributions:** Conceptualization, R.G.; methodology, R.G.; software, H.A. and M.K.; validation, M.K. and R.G.; formal analysis, R.G. and M.K.; investigation, A.O. and H.A.; data curation, R.G., H.A., A.O. and M.K.; writing—original draft preparation, H.A., R.G. and M.K.; writing—review and editing, H.A., R.G. and M.K.; visualization, M.K. and H.A.; supervision, R.G.; project administration, R.G. and A.O.; funding acquisition, A.O. and R.G. All authors have read and agreed to the published version of the manuscript.

**Funding:** The study was financially supported by the project coordination unit of Igdir University (Türkiye), with project number 2018-FBE-L02. In this regard, we would like to send our deep thanks to Igdir University.

**Institutional Review Board Statement:** Not applicable.

**Data Availability Statement:** The data used to support the findings are all included in a Supplementary File.

**Conflicts of Interest:** The authors declare that they have no known competing financial interest or personal relationship that could have appeared to influence the work reported in this paper.

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
