# Peer review of "Management of Weeds in Maize by Sequential or Individual Applications of Pre- and Post-Emergence Herbicides"

_agriculture, doi:10.3390/agriculture13020421_

Round 1

Reviewer 1 Report

Comments on MS agriculture-2180323 entitled “Management of weeds in maize by sequential or individual applications of pre- and post-emergence herbicides”

The study is on efficacy of sequential and individual applications of Dimethenamid-P + Terbuthylazine and Isoxaflutole + Thiencarbazone methyl + Cyprosulfamide as pre- emergence on maize crop.

General comments

English has to be improved in grammar and syntax in the MS

Eg Page 1, line 21 -22 “On the other hand, Mesotrione + Nicosulfuron and Dicamba + Nicosulfuron used  as post- emergence herbicides”

Change to “On the other hand, Mesotrione + Nicosulfuron and Dicamba + Nicosulfuron  was used  as post- emergence herbicides

Page 1 line 22 – 23 “Hereby, the effects of the herbicides were also assayed on corn yield 22 and related parameters.”

Delete the Hereby

The MS has to checked for grammar and syntax errors.

The introduction is well written, it is  concise, informative, and engaging, providing the necessary background information, literature review and context for the study. It also sets the tone of the whole paper, by presenting the main hypothesis, objectives and objectives of the research.

The materials and methods section is well written, it has all the details of the methods in a clear way and the experiment is well planned and well executed. The details on the research design, experimental treatments, sample size, and data collection methods is given clearly.

In the results and discussion section the authors have given all the details of the results obtained in a very clear way.

In the sub heading “Weed species and their density” the authors have to explain the results obtained. They have clearly written the results, they have to add explanations for the results obtained. They have to discuss the reasons for the presence of high and low density of weeds.

In the sub heading “Effects of herbicides on the weed species and population” , the authors have said that “ Some systemic herbicides may become effective within a few days of application, while others may take longer to become effective. The  speed at which a systemic herbicide becomes effective may also depend on environmental factors such as temperature, humidity, and sunlight”. The authors have to explain this with reference to the results obtained in the study.

Overall the results and discussion section has to rewritten with clear and more explanations for the results obtained. The section is mostly results and not much discussion.

The sub headings “Variance analysis of weed dry weight and agronomic attributes” and “Multivariate analysis of the parameters and treatments” is about the methods used and the results are given in  a clear way , the authors have to add discussion for these results.

The conclusion should have more explanations on the results obtained.

The MS can be accepted after major revisions.

Author Response

English language and style: (x) Moderate English changes required

Dear Reviewer-1; the language of the study was double-checked and revised accordingly. The changes were annotated in blue in the manuscript.

Are the results clearly presented?:  (x) Must be improved

Dear Reviewer-1, we totally agree with you. There were some points to be revised. We clarified accordingly. Furthermore, the results in conclusion section was also double-checked and revised.

Are the conclusions supported by the results?: (x) Must be improved

Dear Reviewer-1, the results in conclusion section was also double-checked and revised.

Comments and Suggestions for Authors (Reviewer 1)

General comments

English has to be improved in grammar and syntax in the MS

Dear Reviewer, the manuscript was double-checked for syntax, punctuation and grammar.

Eg Page 1, line 21 -22 “On the other hand, Mesotrione + Nicosulfuron and Dicamba + Nicosulfuron used as post- emergence herbicides”

Change to “On the other hand, Mesotrione + Nicosulfuron and Dicamba + Nicosulfuron was used as post- emergence herbicides

Dear Reviewer, upon your request; we did the relevant change. Thank you very much.

Page 1 line 22 – 23 “Hereby, the effects of the herbicides were also assayed on corn yield and related parameters.”

Delete the Hereby

Dear Reviewer, it was deleted upon your kind request

The MS has to checked for grammar and syntax errors.

Dear Reviewer, the manuscript was double-checked for syntax, punctuation and grammar.

The introduction is well written, it is concise, informative, and engaging, providing the necessary background information, literature review and context for the study. It also sets the tone of the whole paper, by presenting the main hypothesis, objectives and objectives of the research.

Dear Reviewer, we really appreciate your kind words. Thank you very much.

The materials and methods section are well written, it has all the details of the methods in a clear way and the experiment is well planned and well executed. The details on the research design, experimental treatments, sample size, and data collection methods is given clearly.

Dear Reviewer, we really appreciate your kind words. Thank you very much.

In the results and discussion section the authors have given all the details of the results obtained in a very clear way.

Dear Reviewer, we really appreciate your kind words. Thank you very much.

In the sub heading “Weed species and their density” the authors have to explain the results obtained. They have clearly written the results, they have to add explanations for the results obtained. They have to discuss the reasons for the presence of high and low density of weeds.

Dear Reviewer, upon your request; we paraphrased and extended the relevant explanation with new references. It was as follows:

 “As of other living organisms, weeds are also critically responsive to the environmental fluctuations, either in biotic or abiotic nature (Poggio, 2005; Netto, 2022). Regardless of en-vironmental conditions, the wide-spread and density of the weeds might be linked to the characteristics of the species such as competition ability, seed production, number of seeds per plants, dissemination, and reproduction system (annual and perennial) (Poggio, 2005; Netto, 2022).”

In the sub heading “Effects of herbicides on the weed species and population” , the authors have said that “ Some systemic herbicides may become effective within a few days of application, while others may take longer to become effective. The speed at which a systemic herbicide becomes effective may also depend on environmental factors such as temperature, humidity, and sunlight”. The authors have to explain this with reference to the results obtained in the study.

Dear Reviewer, upon your request; we paraphrased and extended the relevant explanation with new references. It was as follows:

The herbicides are classified into two major groups according to their translocation characteristics in plants, contact and systemic herbicides (Mersie and Singh, 1989). As clearly uttered by Cobb (2022) and [54], the time required for systemic herbicides is related to a quite number of factors such as herbicide type, target weed, environmental conditions, and application methods. The effects of systemic herbicides might be manifested from a few hours to several weeks. Also, the final effects might not be observed for several weeks as the plant continues to absorb the herbicide.

Overall the results and discussion section has to rewritten with clear and more explanations for the results obtained. The section is mostly results and not much discussion.

Dear Reviewer, we extended the result and discussion sections with new and recent references.

The sub headings “Variance analysis of weed dry weight and agronomic attributes” and “Multivariate analysis of the parameters and treatments” is about the methods used and the results are given in a clear way, the authors have to add discussion for these results.

Dear Reviewer, we extended the result and discussion sections with new and recent references.

The conclusion should have more explanations on the results obtained.

Dear, we have paraphrased the conclusions with additions. Now, it is clearer.

The MS can be accepted after major revisions.

Dear Reviewer, we would like to thank you very much for your constructive and valuable comments in improving the final version of the manuscript.

Reviewer 2 Report

In this research, The authors are studying the Management of weeds in maize by sequential or individual applications of pre- and post-emergence herbicides. The author designed the present study to compare the individual and combined effects of the herbi cides with the different bioactive compounds. In order to test the hypothesis, a series of parameters such as weed dry weight, grain yield, plant height, kernel rows, cob length, core diameter, and 1000-grain weight were recorded in maize plants. However, there were still some shortcomings:

1. Line 110, 2 in K2O should be the subscript, please check, the same below.

2. Materials and Methods, study area as well as the experiment treatments diagram should be put there.

3. What is the basis for the experimental design of this study ?

4. Figure 3 and Figure 4 should add standard error bars and one-way ANOVA.

5. Figure 6 is not clear.

6. In the third part, there is a lack of comparative analysis between the conclusions of this study and the previous research results.

7. It is suggested to quote more articles in the past three years.

Author Response

Comments and Suggestions for Authors (Reviewer 2)

Are all the cited references relevant to the research? : (x) Can be improved

Dear Reviewer, we extended the manuscript with additional relevant references. Those were annotated in blue in the reference section.

Is the research design appropriate? : (x) Can be improved

Dear Reviewer, upon your kind request; we clarified the experimental design with a diagram. It is presented in Figure 1.

Are the results clearly presented? : (x) Can be improved

Dear Reviewer, we totally agree with you. There were some points to be revised. We clarified accordingly. Furthermore, the results in conclusion section was also double-checked and revised.

Are the conclusions supported by the results? :  (x) Can be improved

Dear Reviewer, the results in conclusion section was also double-checked and revised.

Comments and Suggestions for Authors

In this research, the authors are studying the Management of weeds in maize by sequential or individual applications of pre- and post-emergence herbicides. The author designed the present study to compare the individual and combined effects of the herbi cides with the different bioactive compounds. In order to test the hypothesis, a series of parameters such as weed dry weight, grain yield, plant height, kernel rows, cob length, core diameter, and 1000-grain weight were recorded in maize plants. However, there were still some shortcomings:

Dear Reviewer, we would like to thank you very much for your constructive and leading comments. Those really improved the manuscript. We, all authors, appreciate your comments.

  1. Line 110, 2 in K2O should be the subscript, please check, the same below.

Dear Reviewer, upon your request, they were double-checked and revised accordingly.

  1. Materials and Methods, study area as well as the experiment treatments diagram should be put there.

Dear Reviewer, upon your kind request; we inserted a diagram of experimental design as Figure 1.

  1. What is the basis for the experimental design of this study?

Dear Reviewer; the experiment is based on a randomized complete block design with four replications.

“The experiment consisted of forty plots with ten experimental groups (DT, ITC, MN, DN, DT+MN, DT+DN, ITC+MN, ITC+DN, weed free and weedy) and four replications according to the randomized complete blocks design (Figure 1).”

  1. Figure 3 and Figure 4 should add standard error bars and one-way ANOVA.

Dear Reviewer, we totally agree with you. Upon your request, we did all relevant changes.

  1. Figure 6 is not clear.

Dear Reviewer, the relevant figure is now clear to the readers.

  1. In the third part, there is a lack of comparative analysis between the conclusions of this study and the previous research results.

Dear Reviewer, we extended the relevant sections of the study by additions of recent references.

  1. It is suggested to quote more articles in the past three years.

Dear Reviewer, the manuscript was extended with new references.

Reviewer 3 Report

Manuscript ID: agriculture-2180323-peer-review-v1

Manuscript Title: Management of weeds in maize by sequential or individual applications of pre- and post-emergence herbicides.

The title and subject of the manuscript are very interesting from the methodological and practical point of view, suitable and adequate. The scientific content contributes to the space in which it develops.

The analysis of the published data was provided with a sufficient level of scientific novelty. The abstract of the paper is factual concrete, realistic, and understandable.

The introduction provides a good understanding of the subject and its importance, with a significant quantity of information. Theoretical and practical reasons for the experiments are very reasonable.

The materials and methods are written clearly and in detail for the reader to understand.

The results were described nicely and accurately and discussed very well according to my knowledge.

There are some minor corrections that I have noticed that may improve the standard of the manuscript.

I hope my comments improve the quality of your manuscript

Best regards

Author Response

Comments and Suggestions for Authors (Reviewer 3)

Are all the cited references relevant to the research? : (x) Can be improved

Dear Reviewer-3; we extended the manuscript with new relevant references, the relevant additions were annotated in blue.

Are the methods adequately described? :  (x) Can be improved

The method and its design were clarified by addition of a diagram of experimental design.

Are the results clearly presented? :  (x) Can be improved

Dear Reviewer, we totally agree with you. There were some points to be revised. We clarified accordingly. Furthermore, the results in conclusion section was also double-checked and revised.

Comments and Suggestions for Authors

The title and subject of the manuscript are very interesting from the methodological and practical point of view, suitable and adequate. The scientific content contributes to the space in which it develops

Dear Reviewer, we really appreciate your kind words. Thank you very much.

The analysis of the published data was provided with a sufficient level of scientific novelty. The abstract of the paper is factual concrete, realistic, and understandable.

Dear Reviewer, we really appreciate your kind words. Thank you very much.

The materials and methods are written clearly and in detail for the reader to understand.

Dear Reviewer, we really appreciate your kind words. Thank you very much.

The results were described nicely and accurately and discussed very well according to my knowledge.

Dear Reviewer, we really appreciate your kind words. Thank you very much.

There are some minor corrections that I have noticed that may improve the standard of the

Dear Reviewer, we really appreciate your kind words. Thank you very much.

manuscript.

I hope my comments improve the quality of your manuscript

Dear Reviewer, we did all the changes you have kindly proposed. We would like to thank you very much for your constructive and valuable comments in improving the final version of the manuscript.

Dear Reviewer, you have kindly mentioned to the use order of CODES in Table 2 and 6. We completely agree with you but in Table-6; we calculated the effect (%) of the herbicides in comparison to control (weed-free). So, we inserted “control” group at the top of the table to be clearer to the readers. Dear Reviewer, we also noted your concern of the figure by inserting the definition of legends. Also, you have kindly asked your concern about Figure 5-6-7; as we also noted in the text.  The figure was composed of average values of both years. In addition; Abbott is the name of the author who cited as (49).

Round 2

Reviewer 1 Report

The authors have corrected the MS according to the reviewer's suggestions. The MS can be accepted.